# Factors associated with parent-teacher hyperactivity/inattention screening discrepancy: Findings from a UK national sample

**Hei Ka Chan** **\*, Richard Rowe, Daniel Carroll**

Department of Psychology, University of Sheffield, Sheffield, United Kingdom

\* hkchan3@sheffield.ac.uk, nadia.heika.chan@gmail.com

## Abstract

### Background

To fulfil the diagnostic criteria of Attention Deficit Hyperactivity Disorder in the Fifth Edition of Diagnostic and Statistical Manual of Mental Disorders (DSM-5), symptoms should be observed in two or more settings. This implies that diagnostic procedures require observations reported from informants in different settings, such as teachers in school and caregivers at home. This study examined parent-teacher agreement in reporting hyperactivity/inattention and its relationship with child's, parent's, and family's characteristics.

### Method

We used data from the 2004 United Kingdom Mental Health of Children and Young People survey, including 7977 children aged 4–17, to investigate cross-informant agreement between parents and teachers on the hyperactivity-inattention subscale of the Strengths and Difficulties Questionnaire. The characteristics of different patterns of informant agreement were assessed using multinomial logistic regression.

### Results

Cross-informant agreement of parent and teacher was low (weighted kappa = .34, 95% C.I.: .31, .37). Some characteristics, such as male child and parental emotional distress, were associated with higher likelihood of parent-teacher discrepancy.

### Conclusion

We found low informant agreement in the hyperactive/inattention subscale, as hypothesised and consistent with previous studies. The current study has found several factors that predict discrepancy, which were partly consistent with previous research. Possible explanation, implications, and further research on parent-teacher informant discrepancy in reporting hyperactivity/inattention were discussed.

**Data Availability Statement:** Data cannot be shared publicly because of access level designed by the UK Data Service. Data are available from the UK Data Service (contact via https://ukdataservice.

ac.uk/) for researchers who meet the criteria for access to confidential data. The data underlying the results presented in the study are available from https://beta.ukdataservice.ac.uk/datacatalogue/studies/study?id=5269.

**Funding:** The author(s) received no specific funding for this work.

**Competing interests:** The authors have declared that no competing interests exist.

## Introduction

Attention deficit hyperactivity disorder (ADHD), a disorder with features of developmentally inappropriate levels of hyperactivity, impulsivity and/or inattention, affects both children and adults worldwide [1–3]. Identifying ADHD is essential for accessing interventions for improving functioning [4], minimising possible longer-term consequences [5], and reducing poorer outcomes from undiagnosed and/or untreated ADHD [6]. To assess children's and adolescent's problems, parents and teachers are key informants to provide information.

Parent-teacher rating discrepancy is one of the common dilemmas that clinicians face in practice. This is no surprise to observe low to moderate levels of correlations among informants in all forms of paediatric psychopathology [7–9]. There is no exception for hyperactivity/inattention problems for this phenomenon regardless of tools that applied: parent-teacher agreement correlations are usually low-to-moderate. Estimates range from 0.34 to 0.64 [10] using the Strengths and Difficulties Questionnaire's [SDQ; 11] hyperactivity/inattention subscale and from 0.17 to 0.60 [12] using the DSM-attentional problems using the Child Behavior Checklist ratings [13]. Understanding the reasons for informant-discrepancy informs evidence-based practice for both researchers and practitioners, facilitating the integration and interpretation of different informant ratings when screening for psychiatric disorder [14]. Several factors have been investigated in previous research as follows:

### Child's characteristics

**Age.** ADHD symptom presentation changes across development, with declining hyperactivity-impulsivity as age increases, while inattention remains relatively stable [15–18]. Age effects are found in both parent and teacher ratings of hyperactivity/impulsivity, in which higher ratings are observed in younger children by both informants [19, 20]. Murray and colleagues [21] found that parent-teacher discrepancy was independent of child's age and that there is an opposite trend in inattention along age of child, with parent-reported symptoms increasing while teacher-reported symptoms decrease. Meanwhile, Chamorro and colleagues [22] found a similar trend that teachers report less ADHD symptoms as children develop whereas parents-reported symptoms remained stable, inducing an increase in the discrepancy in later childhood. Overall, studies are inconsistent in their results regarding the effect of age on informant agreement of children's ADHD behaviours and parent-teacher agreement has been found to be unrelated to age in some studies [19, 23–26].

**Gender.** ADHD has been reliably found to be more common in males [27–30]. Gender biases on assessing behaviours might affect ratings by informants even when similar problem behaviours are displayed in the same setting [31]. Research conducted by Chamorro and colleagues on 789 Mexican students from six elementary school grades reveals that the discrepancy regarding hyperactivity may be greatest in girls [22]. However, other studies have found no gender difference in parent-teacher agreement [23, 26, 32].

### Parent-rater's characteristics

**Gender.** Fathers tend to identify more ADHD symptoms in screening assessments than mothers [33]. Regardless of moderate mother-father agreement in ADHD [34], there is little evidence available regarding whether the parent gender will affect parent-teacher discrepancy. Some studies have found parental gender is not related to parent-teacher agreement in ADHD [34, 35]. Consistently, reporting genders or roles of caregivers would be helpful in improving such analysis on the gender-related impacts on discrepancies.

**Age.** Previous studies found inconsistent results regarding the relationship between parent age and parent-teacher agreement in reporting child's behaviours [36, 37]. Stone and

colleagues [36] found lower parent-teacher agreement with young mothers only in externalising behaviours in samples of 5 to 12 year-old children, while Munzer and colleagues [37] observed higher parent-teacher agreement with young parents and lower parent-teacher agreement with older parents in both internalising and externalising behaviours in pre-schoolers. However, there is limited existing research exploring how parent age may relate to parent-teacher agreement in ADHD.

**Education.**  Parent-teacher disagreement could result from differences in perception under the influence of parent's characteristics [36]. Education level is associated with awareness of, knowledge about and acceptance of treatment for ADHD [38–40]. Parents with higher education levels may obtain more accurate information about ADHD providing a greater awareness and knowledge of ADHD [38, 40]. Regarding the effect of received education by parents on parent-teacher agreement in ADHD, limited evidence is available and the result is also inconclusive [23, 41]. Parents with higher education predicted more severe reports of inattention than teachers [41]. Parents with higher education levels were most accurate in predicting ADHD diagnosis compared with teachers, and parents who received less education were the least accurate [42]. A relationship between parental education level and parent-teacher disagreement in ADHD was observed in some studies [41, 42]. However, other evidence indicates that parental education level was was unrelated to parent-teacher agreement in both community and clinical samples [23].

**Mental health.**  The depression-distortion hypothesis suggests that when a parent suffers from depression, their rating of their child's behaviours would be less accurate and hence affect cross-informant agreement [43]. Results from Harvey and colleagues [32] supported this hypothesis finding that mothers, but not fathers, with depression were more likely to rate child attention problems more highly than teachers. However, comparing mothers with and without depression, Madsen and colleagues [44] found that mother-teacher disagreements were reduced among mothers with depression. Instead of parental depression, van der Oord and colleagues [45] observed that when a parent reported more parenting stress, parent-teacher discrepancy in rating ADHD symptoms was greater. Higher parenting stress predicted more reported ADHD symptoms by mother than teacher [41]. Results from Chen and colleagues [46] supported the findings of association between parenting stress and informant disagreement from van der Oord and colleagues [45] and Yeguez and Sibley (41). The current findings are not conclusive on the influence of parental depression, parenting stress and parental ADHD on cross-informant agreement, especially on the mechanism of how psychological well-being might influence the agreement. Other than reporting bias from the parent, it is also possible that a parent with depression or parenting stress may be more likely to encounter more behavioural problems at home and more impaired parent-child interaction [47].

## Family's characteristics

**Ethnicity.**  Cultural diversity affects attitudes towards ADHD, knowledge of ADHD, and help-seeking behaviours [48]. Children in ethnic minorities are also less likely to receive ADHD diagnosis and treatment [49]. An association between ethnicity of parents and parent-teacher disagreement in ADHD has been observed in previous work [50]. Parents from ethnic minority were more likely to rate inattention and hyperactivity lower than teachers [32, 50, 51], while Latina mothers were more likely to rate hyperactivity higher than teachers [32]. Compared to non-Latino, teachers were more likely to identify symptoms in Latino youths while parents were less likely to identify symptoms [52]. However, these patterns have not been found consistently. Another study found that parents identified more symptoms than teachers but that the disagreements between parents and teachers were not related to the

ethnicity [53]. Although some studies are available, the literature is not yet sufficiently developed to conclude on how informant agreement in ADHD varies across cultures [23, 32, 50, 54].

**Socioeconomic Status (SES).** Children in lower income families are more at risk for ADHD diagnosis and medication use [30]. Lawson and colleagues [51] found lower SES leads to higher parent-teacher discreprency. There was a higher chance of disagreement regarding inattention symptopms and better agreement on hyperactivity/impulsivity symptoms. However, Takeda et al. [50] found no significant relationship between SES and parent-teacher agreement despite using the same measure of SES as Lawson and colleagues [51]. Saffer et al. [23] also found household income was unrelated to parent-teacher discrepancy.

**Family structure.** The structure of a family, such as numbers of children in the household and parental partnership status may affect how an informant perceives and rates a child's behaviours. For externalising problems, mother-teacher agreement was the best in the single child group and discepancy was greater when more siblings were present [55]. The authors suggested that increased family size might create more parenting stress and lower tolerance for behavioral problems, which increase the chance to rate more symptoms. However, Harvey and colleauges [32] did not find support for the hypothesis on the relationship between number of children in a household with parent-teacher agreement in ADHD. Single parents experienced more parenting stress with a child with ADHD [56] which may affect their rating. However, marital status was not associated with parent-teacher agreement in ADHD [41].

Limited evidence is currently available for a solid conclusion about the relationship of child, parent, and family factors with parent-teacher informant agreement in ADHD. In the present study, we explored how the factors discussed above were associated with parent-teacher agreement/disagreement in ADHD screening. We examined the correlations of parent and teacher reports on the hyperactivity scale of the SDQ [11] in the United Kingdom's 2004 survey of Mental Health of Children and Young People (MHCYP) [57]. We expected to find a low correlation between reporters, following Vaz and colleagues [58] who found low agreement [weighted kappa = .31, 95% confidence interval (C.I.): .13, .48]. Second, we examined the effects of characteristics of the parent, child, and family on parent-teacher agreement. We expected to identify relationships between the informant agreement and the selected factors, such as age, gender, ethnicity, family structure, socioeconomic characteristics, and parent's mental health condition. For instance, we predicted that parents with lower education may identify fewer problems, even if the child presents with inattention-hyperactivity symptoms, which will result in discrepancies with teacher reports.

## Methodology

### Sample and data collection

The data was taken from the 2004 MHCYP survey in the United Kingdom [57]. Details of this survey can be found at Green and colleagues [57]. The data consisted of multiple informants reporting on children aged 4 to 17. A sample of 12,294 families was identified from the Child Benefit Register, and 10,496 (85% of the sample) families were invited to participate in the interview. Fifteen percent of identified families were not interviewed due to opt-outs, moving without trace, and ineligibility. Parents and children aged from 11 to 17 completed the interview and answered a self-rated questionnaire. Consent from parents was sought to contact a nominated teacher who was then invited to participate in the study.

| | Teacher | |
|---|---|---|
| Parent | "Normal" | "Borderline" or "Abnormal" |
| "Normal" | Both agreed not at risk | Teacher-reported only |
| "Borderline" or "Abnormal" | Parent-reported only | Both agreed at risk |

**Fig 1. Categorization for parent-teacher agreement/disagreement patterns on the hyperactivity-inattention subscale of the Strengths and Difficulties Questionnaire.**

## Missing data

In the existing dataset, 7977 families achieved complete interviews with data from up to three reporters. Ninety four percent of families provided parental consent for teachers' participation and 6236 (78% of all interviews) teachers provided a response. In the current study, any participant providing one missing or unrecognised coding was excluded. Pairwise deletion was used in the current analysis. This meant a sample of 5781 children with both parent- and teacher-reported ratings remained, after data cleaning.

## Measures

**Hyperactivity-inattention.** The SDQ [11] is a 25-item brief informant questionnaire for screening psychopathology of children from 4–17 over the past six months and available for parents, teachers, and children over 11 years old. The hyperactivity-inattention subscale contains 5 items addressing restlessness ("*restless, overactive*" and "*constantly fidgeting or squirming*"), distraction ("*easily distracted, concentration wanders*"), and impulsivity ("*thinks things out before acting*" and "*sees tasks through to the end*"). It has been shown to be a useful screening tool for ADHD [e.g. 59, 60]. The items were scored as "*not true = 0*", "*partly true = 1*" and "*certainly true = 2*", with two items reverse coded. The sum of the five items of this subscale from all informants can be categorised into "*normal (0–5)*", "*borderline (6)*" and "*abnormal (7–10)*" [11]. The categorisation instead of the total scores of the subscale were used for comparisons. The current study proposes a categorical variable set for informant patterns. using the SDQ band category system [11], the "*borderline (6)*" and "*abnormal (7–10)*" were regrouped in one group ("*at risk*"), which is consistent with previous approaches [58], and formed a 2x2 matrix of parent-teacher agreement/disagreement patterns as displayed in Fig 1.

## Background measures

Demographic variables, such as age, gender ['female' (0), 'male' (1)], ethnicity ['White' (0), 'Non-White' (1)], family structure ['not lone parent' (0) or 'lone parent' (1), and number of children in household], socioeconomic characteristics ['above national average income' (0) or 'below national average income' (1) [61]], family's employment status ['both parents or one parent working' (0) or 'neither parent working' (1)], parents' educational level ['completed at least a formal degree' (0) or 'did not completed any formal degree' (1)]. The last variable in the current study was the parent's self-reported mental health condition which was measured using the 12-item General Health Questionnaire [GHQ; 62]. The GHQ items were scored as "more so than usual or same as usual (0)" and "less so than usual or much less than usual (1)".

A total score ranging 0 to 12 was calculated and grouped into "screened positive for an emotional issue (1)" for a total score of 3 or more and "not screened positive for an emotional issue (0)", as defined by Green et al. [57].

## Ethical considerations

The MHCYP study was conducted by the Office for National Statistics and commissioned by the Department of Health and the Scottish Executive Health Department in the United Kingdom. Written consent was obtained from parents and verbal consent was obtained from children before participation. Consents from parents were sought to contact the nominated teacher for providing data. The pre-analysis dataset was fully anonymized at the data archive and was downloaded from https://beta.ukdataservice.ac.uk/datacatalogue/studies/study?id=5269 at 24 April 2021. Ethics approval (Reference Number: 040880) for the current analyses was obtained from the Ethics Committee at the University of Sheffield, adhering to 'Research Ethics: General Principles and Statements' for secondary analysis.

## Statistical analysis

STATA17 [63] was used for all analyses. Cross-informant correlation on the hyperactivity-inattention subscale was examined using weighted kappa. Multinomial logistic regression was conducted to explore predictors of parent-teacher agreement/disagreement patterns. Tests of assumptions [variance information factor (VIF) for multicollinearity between variables, Hosmer-Lemeshow (H-L) statistic for homogeneity of multinomial logistic regression] of the regression analyses were also conducted. Wald tests on coefficients were employed to test the impact of each factor on likelihood of falling into different agreement/disagreement groups.

# Results

## Sample description

The dataset contained 7977 parent/child dyads. The majority of parent-informants were female (94.40%) and approximately half of the sampled children were male (51.54%). Child ages ranged from 4 to 17 with a mean age 10.54 years (SD = 3.40). The age of the interviewed parent ranged from 18 to 78 years old (M = 39.08, SD = 6.43). Ethnic origin of White (88.22%) and non-White (11.78%) are recorded. In terms of family structure, the percentage of lone parents was 24.29% and the mean number of children in a household was 2.13 (SD = 1.07). Regarding annual household income, 51.02% of families were below national average income. Regarding the family's employment status, the percentage of neither parent working was 15.42. The percentage of interviewed parents reported completing any educational qualifications was 81.82%. In terms of emotional issues measured by GHQ-12, 22.58% of interviewed parents scored as "screened positive". Most children fell into the not at-risk range on the hyperactivity/inattention subscale of the SDQ. Cross-informant agreement on the hyperactivity-inattention subscale was low (weighted kappa = .34, 95% C.I.: .31, .37; see Table 1).

## Characteristics of parent-teacher discrepancy

Four categories were formed according to the parent-teacher agreement/disagreement patterns: both-agreed not at risk, parent-reported at risk only, teacher-reported only, and both-agreed at risk, as shown in Table 1. Multinomial logistic regression (LR chi2 = 558.58, p < .001, pseudo R2 = .060) was used to regress the categorical outcome variable onto child (gender, age); parent (gender, age, emotional issue, and education level); and family characteristics (marital status of parents, employment, household income, number of children in household,

**Table 1. Frequency of categorization for parent-teacher agreement/disagreement patterns on the hyperactivity-inattention subscale of the Strengths and Difficulties Questionnaire.**

| Teacher | Not at-risk | At-risk |
|---|---|---|
| Parent | N = 4931 | N = 997 |
| Not at-risk | Both agreed not at-risk | Teacher-reported only |
| N = 6265 | n = 4201 (72.67%) | n = 508 (8.79%) |
| At-risk | Parent-reported only | Both agreed at risk |
| N = 1494 | n = 607 (10.50%) | n = 465 (8.04%) |

and ethnicity). Descriptive statistics and relative risk ratios are displayed in Table 2, including the distribution of the four parent-teacher agreement/disagreement patterns. Wald tests comparing the informant-agreement categories on each factor are shown in Table 3.

In terms of child's characteristics, male children were more likely to be rated as at risk by parent alone (RRR = 1.93, p < .001, C.I.: 1.61, 2.32) and by teacher alone (RRR = 3.29, p < .001, C.I.: 2.65, 4.08), comparing to when both informants rated as not at-risk. Male children were less likely to be rated by at risk by parent-alone compared to teacher-alone (chi2 = 15.22, p < .001) and by both informants (chi2 = 23.59, p < .001) as at risk. Child' age was not related to parent-teacher discrepancy in the current study.

In terms of parent's characteristics, children rated by older parents were found to be less likely of being rated as at-risk by parent alone (RRR = .98, p < .01, C.I.: .96, .99) and by teacher alone (RRR = .98, p < .05, C.I.: .96, 1.00), compared to when both informants rated as not at-risk. Parents who are screened as positive for an emotional issue were more likely to rate their children at risk by parent alone (RRR = 1.62, p < .001, C.I.: 1.32, 1.99) compared to when both informants reported the child as not at risk. We also found that when parents with positively screened emotional issues, their children were more likely to be rated by parent-alone (chi2 = 5.51, p < .05) and by both informants (chi2 = 7.69, p < .01) than by teacher alone as at risk. For parents without a formal degree, children were more likely to be rated by parent alone (RRR = 1.69, p < .01, C.I.: 1.24, 2.29) as at risk, compared to when both informants rated as not at risk. They were also more likely to be rated as at risk by both informants than being rated by teacher-only (chi2 = 6.76, p < .01). Parental gender was not related to parent-teacher discrepancy.

In terms of family's characteristics, children from families where neither parent was working were more likely to be rated at risk by teacher only (RRR = 1.61, p < .01, C.I.: 1.18, 2.19) in comparison to both informants rated as not at risk. They were slightly more likely to be rated by both informants (chi2 = 3.91, p < .05) as at risk than by parent only. Family with below average income (RRR = 1.33, p<0.01, C.I.: 1.08, 1.65) and parents who identified themselves as White (RRR = 0.60, p<0.05, C.I.: 0.41, 0.88) were more likely to rate their children at risk by parent alone, when compared to both informants rated as not at-risk. Our results also found that non-White children were more likely to be rated by teacher-alone as at risk compared with parent-alone (chi2 = 8.89, p < .01) and both informants (chi2 = 10.60, p < .01) as at risk. Parental marital status and number of children in the household were not associated with discrepancy in the current study.

## Discussion

### Cross-informant correlations

As hypothesised on the basis of previous research, we found a weak to moderate correlation between informants' hyperactivity ratings that was consistent with the relationships found in

**Table 2. Risk factors distributed among parent-teacher agreement/disagreement patterns and parent-teacher agreement/disagreement patterns predicted by risk factors.**

| Risk factors | Mean/Rate[a] | | | | RRRs (base: Both agreed not at risk) | | |
|---|---|---|---|---|---|---|---|
| | Both agreed not at risk (n[b] = 4201) | Parent-only (n[b] = 607) | Teacher-only (n[b] = 508) | Both agreed at risk (n[b] = 465) | Parent -only | Teacher-only | Both agreed at risk |
| Child characteristics | | | | | | | |
| Male | 44.80% | 59.97% | 73.23% | 75.05% | **1.93\*\*\* (1.61, 2.32)** | **3.29\*\*\* (2.65, 4.08)** | **3.87\*\*\* (3.06, 4.89)** |
| Age | 10.43 | 9.92 | 9.91 | 9.68 | 0.97 (0.94, 1.00) | 0.97 (0.94, 1.01) | **0.96\* (0.93, 1.00)** |
| Parent characteristics | | | | | | | |
| Male | 4.48% | 4.94% | 7.09% | 3.44% | 1.17 (0.75, 1.82) | 1.46 (0.97, 2.22) | 0.78 (0.43, 1.40) |
| Age | 39.31 | 37.81 | 38.13 | 36.88 | **0.98\*\* (0.96, 0.99)** | **0.98\* (0.96, 1.00)** | **0.96\*\*\* (0.95, 0.98)** |
| Positive for emotional issue | 19.99% | 29.57% | 22.85% | 33.19% | **1.62\*\*\* (1.32, 1.99)** | 1.15 (0.90, 1.46) | **1.77\*\*\* (1.40, 2.23)** |
| Without a formal degree | 14.84% | 18.67% | 22.66% | 28.51% | **1.69\*\* (1.24, 2.29)** | 1.22 (0.90, 1.65) | **2.41\*\*\* (1.55, 3.75)** |
| Family characteristics | | | | | | | |
| Lone parent | 20.73% | 25.04% | 28.35% | 34.84% | 0.89 (0.69, 1.15) | 1.00 (0.76, 1.32) | 1.09 (0.83, 1.45) |
| Neither parent working | 11.32% | 16.45% | 20.28% | 27.65% | 1.09 (0.81, 1.48) | **1.61\*\* (1.18, 2.19)** | **1.61\*\* (1.19, 2.18)** |
| Below average income | 44.54% | 55.26% | 56.28% | 67.70% | **1.33\*\* (1.08, 1.65)** | 1.27 (0.99, 1.61) | **1.73\*\*\* (1.33, 2.24)** |
| Number of children in household | 2.12 | 2.19 | 2.20 | 2.22 | 1.01 (0.92, 1.11) | 0.98 (0.89, 1.09) | 0.98 (0.88, 1.10) |
| Ethnicity as Non-White | 10.93% | 9.72% | 14.2% | 7.96% | **0.60\* (0.41, 0.88)** | 1.22 (0.88, 1.69) | **0.50\*\* (0.31, 0.80)** |

RRR, relative risk ratio; C.I., confidence interval.Bold figures indicate statistically significant findings:

\*p < .05;

\*\*p < .01;

\*\*\*p < .001.

[a]All numbers correspond to percentages, except for age given in mean number of years and numbers of children in household.

[b]Ns vary slightly for each risk factor due to occasional missing data.

all forms of paediatric psychopathology [e.g., 7–9, 64]. Similar findings of low parent-teacher correlations on the SDQ hyperactive-inattention subscale were observed in previous research in both clinical [e.g. 65] and community [e.g., 66] samples.

## Characteristics of parent-teacher discrepancy

The present study also explored relationships among child, parent, and family characteristics with the informant agreement/disagreement patterns. The current study considered a pool of covariates in analysis simultaneously to jointly distinguish the marginal effects of the factors. Some of the relationships found were consistent with the previous literature. Our findings are partly consistent with previous research on factors not significantly associated with parent-teacher agreement on reporting ADHD symptoms, such as child's age [19, 23–26], parental gender [34, 35], number of children in a household [32] and marital status [41].

**Table 3. Wald test between RRR of informant pairs.**

|  | Parent vs Teacher | Parent vs Both | Teacher vs Both |
|---|---|---|---|
| Child characteristics |  |  |  |
| Male | **15.22**\*\*\* | **23.59**\*\*\* | 1.12 |
| Age | 0.06 | 0.11 | 0.30 |
| Parent characteristics |  |  |  |
| Male | 0.64 | 1.32 | 3.39 |
| Age | 0.01 | 0.86 | 0.96 |
| Positive for emotional issue | **5.51**\* | 0.35 | **7.69**\*\* |
| Without a formal degree | 2.45 | 1.83 | **6.76**\*\* |
| Family characteristics |  |  |  |
| Lone parent | 0.48 | 1.40 | 0.22 |
| Neither parent working | 3.72 | **3.91**\* | 0.00 |
| Below average income | 0.12 | 2.55 | 3.28 |
| Number of children in household | 0.18 | 0.16 | 0.00 |
| Ethnicity as Non-White | **8.89**\*\* | 0.36 | **10.60**\*\* |

Bold figures indicate statistically significant findings:

\*p < .05;

\*\*p < .01;

\*\*\*p < .001.

Regarding child's characteristics, male children were more likely to fall into parent-teacher disagreement in our study, which is consistent with previous findings for externalizing disorders [e.g., 10]. It is also consistent with previous research about gender features of ADHD where it has classically been found that symptoms and diagnosis are more common in males, whether in single-informant or both-informant ratings [e.g. 67]. However, it is noticeable that parent-only screen-positive is less likely than teacher-only and both-informant rating among male children. This may imply the interaction of gender with informant agreement, for example, teacher-informants may have a higher likelihood to overestimate the incident among male children.

Regarding parental characteristics, younger parent-informants were more likely to fall into parent-teacher disagreement. This is consistent with Cheng and colleagues' [10] finding that younger mother-informants were associated with externalizing problems measured by the SDQ. The current result is parallel with previous studies on younger parental age as a risk factor for informant discrepancy when screening ADHD and other externalizing problems in children [36, 37]. In addition, our results showed that parent-teacher disagreement was more common when parents were screen-positive for facing emotional issues. Harvey et al. [32] reported similar findings regarding the association between maternal depression and mother-teacher disagreement; mothers with depression reported more attention problems in their children than teachers. Our findings were consistent with the depression-distortion hypothesis [43], indicating that parents with psychological distress tend to rate more symptoms and are more likely to screen positive alone or to agree with teacher that the child screens positive.

An association between parental education level and parent-teacher disagreement was observed in our study. This is consistent with previous research that has found a positive correlation between parental education levels and knowledge of ADHD [38, 40] and a positive effect of parental education on ADHD identification [42]. Parents with higher education are

suggested to have higher knowledge regarding ADHD are more likely to report higher ADHD symptoms of their children.

Regarding family characteristics, some measures of SES, such as number of working parents, and average household income, were significantly associated with parent-teacher disagreement in reporting ADHD symptoms. Our findings suggested that children from families with below average income were rated to have less symptoms by teachers, which is contradictory to previous findings. For example, in one study of pre-school children [51], both parents and teachers report more symptoms for children with lower SES. This contrast may be due to different measures of SES in the current study (income only) or another unexplained mechanism. Difference in measures of SES may have hinder comparison of current results with previous studies.

Our findings also suggested that parents who identified their ethnicity as Non-White rated lower hyperactivity than teachers, which is consistent with previous studies [32, 50, 51]. Our findings also suggested that children rated by Non-White parent-informants were more likely to be rated as at risk for ADHD by teacher-alone. This finding is similar to a previous study comparing Non-Latino and Latino youths, which found that Latino youths were rated for more symptoms by teachers [52]. However, the results might not be directly comparable, as the classification of ethnicity in this previous study is Latino-oriented. This might suggest the phenomenon appearing across different minor ethnicities. Parental marital status and number of children in the household were not associated with discrepancy in the current study.

The current study has brought insights to both scientific and practical issues. For example, when disagreement in rating occurs, clinicians might consider the factors identified to predict interrater discrepancy, such as parental age and mental status. The current study utilized a large, nationally representative sample, that measures many factors that have been studied as predictors of parent/teacher discrepancy. However, interpretation of our results must be considered in the light of some limitations. There are other possible characteristics that were not included in the current dataset that might contribute to parent-teacher discrepancies. For example, teacher's characteristics, such as ethnicity and stress level, were not measured. Our analysis might have benefited from including such information for exploring possible linkage of informant characteristics with informant agreement/disagreement, as suggested by previous study [e.g. 68]. Nevertheless, our findings confirm results from the previous literature. Future research should focus on enhancing further understanding on the relationships between informant discrepancy with ADHD diagnosis for exploring the impact of agreement/disagreement on screening accuracy.

## Conclusion

We found low informant agreement in the hyperactive/inattention subscale, consistent with previous studies. Several characteristics, such as gender of child and parental age, were associated with parent-teacher agreement/disagreement patterns. In conclusion, the current research has found several factors that played a role in informant rating discrepancy, and thus may be important in interpreting screening results especially when informants disagree with each other. As ADHD diagnosis requires symptoms to present in two or more settings, it is important to explore how the informant discrepancy induced then could have influenced the diagnosis, for example, how actual diagnosis might be different between parent-only and teacher-only screening scenarios, and the impacts of screening accuracy. Further research evaluating the screening accuracy from the informants, and the utilization of all the information attainable in clinical process, to facilitate the usage of informant rating in aiding clinical diagnosis may be an important next step.

## Author Contributions

**Formal analysis:** Hei Ka Chan.

**Methodology:** Hei Ka Chan.

**Supervision:** Richard Rowe, Daniel Carroll.

**Writing – original draft:** Hei Ka Chan.

**Writing – review & editing:** Richard Rowe, Daniel Carroll.

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
