## [Decision Letter · Decision Letter 0]

2 Jan 2024

PONE-D-23-24488Factors associated with parent-teacher hyperactivity/inattention screening discrepancy: Findings from a UK national samplePLOS ONE

Dear Dr. Chan,

Thank you for submitting your manuscript to PLOS ONE. After careful consideration, we feel that it has merit but does not fully meet PLOS ONE’s publication criteria as it currently stands. Therefore, we invite you to submit a revised version of the manuscript that addresses the points raised during the review process. These are described below. Address every point raised by the reviewer and indicate how the manuscript has been amended to reflect the changes.

Please submit your revised manuscript by Feb 16 2024 11:59PM.  If you will need more time than this to complete your revisions, please reply to this message or contact the journal office at plosone@plos.org. Please include the following items when submitting your revised manuscript:A rebuttal letter that responds to each point raised by the academic editor and reviewer(s). You should upload this letter as a separate file labeled 'Response to Reviewers'.A marked-up copy of your manuscript that highlights changes made to the original version. You should upload this as a separate file labeled 'Revised Manuscript with Track Changes'.An unmarked version of your revised paper without tracked changes. You should upload this as a separate file labeled 'Manuscript'.

We look forward to receiving your revised manuscript.

Kind regards,

Gerard Hutchinson, MD

Academic Editor

PLOS ONE

Journal Requirements:

Reviewers' comments:

Reviewer's Responses to Questions

**Comments to the Author**

1. Is the manuscript technically sound, and do the data support the conclusions?

Reviewer #1: Yes

2. Has the statistical analysis been performed appropriately and rigorously? 

Reviewer #1: Yes

3. Have the authors made all data underlying the findings in their manuscript fully available?

Reviewer #1: Yes

4. Is the manuscript presented in an intelligible fashion and written in standard English?

Reviewer #1: Yes

5. Review Comments to the Author

Reviewer #1: In this study, the authors examine the influence of multiple factors (pertaining to child’s characteristics, parent-rater’s characteristics and family’s characteristics) on cross-informant agreement between parents and teachers on the hyperactivity-inattention subscale of the Strengths and Difficulties Questionnaire. To this end, a multinomial regression analysis was performed using a large representative sample (n = 7977) of the UK Mental Health of Children and Young People survey.

Results showed that overall cross-informant agreement of parent and teacher was low. Moreover, concerning child’s characteristics only the child’s sex (but not age) was associated with higher likelihood of parent-teacher discrepancy. With respect to parent-rater’s characteristics, age, mental health issues and educational level were significantly associated with rater disagreement. Among the family characteristics, SES, number of working parents, and ethnicity emerged as significant predictors in the analysis.

General comments

The authors provide a concise but systematic overview of the literature concerned with the factors influencing parent-rater disagreement in ADHD symptoms. The relevance of the study is clearly stated and straightforward. I agree with the authors that there is a need of studies examining the potential influence of the multiple factors simultaneously in one and the same analysis - as done by the authors. I think the study would attract researchers and practitioners in the field, but the manuscript should be improved in some respects. I therefore recommend some minor revisions.

(1) In the theory section, the findings on the factors associated with parent-teacher agreement are presented very systematically, but sometimes the literature is presented too briefly. For instance, it is stated that hyperactivity-impulsivity symptoms decrease with age (p.4) – whereas this is true for the hyperactivity symptoms, there is evidence to suggest that impulsivity and risk behavior increase in adolescents with ADHD compared to children with ADHD. Likewise, it is true that ADHD has been reliably reported more commonly in males (p. 4). Yet, I would appreciate it if the authors could go into more detail about the underlying mechanisms contributing to this gender bias (e.g., different symptom expressions in boys and girls contribute to under-identification of ADHD in girl. The inattantive subtype of ADHD is more prevalent in girls, whereas the combined subtype is more prevalent in boys).

(2) I think it is also important to point out that the existence of a discrepancy between observers says nothing about the accuracy of their respective judgments. Because both informants assess the child's separate environments (home versus school), some of the observed behavioral discrepancy may well reflect true differences in the child's behavior in the two contexts instead of reflecting an informant bias. That is, statements about the accuracy of the respective informants can only be made based on validation with a clinical interview - which, unfortunately, is rarely done in research. I would appreciate if the authors could add a brief overview of this issue in the theory section (e.g., this issue is well presented in the papers by Los Reyes et al., 2015, 2019). Likewise, it should also be mentioned in the discussion that that the present findings do not inform about rater accuracy.

(3) On p. 8, the authors present their hypotheses and “predicted that parents with lower education may identify fewer problems, even if the child presents with inattention-hyperactivity symptoms, which will result in discrepancies with teacher reports”. I think, this hypothesis cannot be referred from the present data. Since no clinical interview was additionally performed with the children, information on the actual inattention-hyperactivity symptoms of the children is missing so we do not know if the child presents with respective symptoms or not. The authors should thus rephrase the sentence.

(4) Although the statistical analysis is appropriate for the question at hand, I wonder whether carrying out a comparable analysis on a latent (instead of on a manifest) level would be possible given the large sample size. (However, I am not an expert in this field, so I am not sure whether those methods exist).

(5) In the discussion, I would like the authors to elaborate on the effect sizes of the respective characteristics. The discussion summarizes the results well, makes appropriate comparisons with previous research, and addresses possible reasons for inconsistent findings from other studies. What I still miss in the discussion is that the authors elaborate on the effect sizes of the significant results – especially given the large sample. For example, it can be seen from Tables 2 and 3 that some of the significant factors examined only show small effects on parent-teacher agreement, while others show larger effects. In my opinion, adding a few sentences on the relevance of the respective significant characteristics would strengthen the discussion.

References

Los Reyes A de, Augenstein TM, Wang M et al. (2015) The validity of the multi-informant approach to assessing child and adolescent mental health. Psychol Bull, 141:858–900. https://doi.org/10.1037/a0038498 521

Los Reyes A de, Cook CR, Gresham FM et al. (2019) Informant discrepancies in assessments of psychosocial functioning in school-based services and research: Review and directions for future research. J Sch Psychol, 523 74:74–89. https://doi.org/10.1016/j.jsp.2019.0

6. PLOS authors have the option to publish the peer review history of their article (what does this mean?). If published, this will include your full peer review and any attached files.

Reviewer #1: No

---

## [Author Response · Author response to Decision Letter 0]

2 Feb 2024

Manuscript PONE-D-23-24488

Response to Reviewer 

Date: 2024/02/01

Dear Dr. Hutchinson,

Thank you for giving us the opportunity to submit a revised draft of the manuscript “Factors associated with parent-teacher hyperactivity/inattention screening discrepancy: Findings from a UK national sample” for publication in the PLOS ONE. We appreciate the time and effort dedicating to providing feedback on our manuscript. We are grateful for the insightful comments and that have led to valuable improvements to our paper. 

Revisions made to the manuscript in response to comments are highlighted with tracked changes. Please see below, in blue, for a point-by-point response to the reviewers’ comments and concerns, and in red, for the changes made. All page numbers refer to the revised manuscript file with tracked changes. 

Reviewer' Comments to the Authors: 

Comment 1: In the theory section, the findings on the factors associated with parent-teacher agreement are presented very systematically, but sometimes the literature is presented too briefly. For instance, it is stated that hyperactivity-impulsivity symptoms decrease with age (p.4) – whereas this is true for the hyperactivity symptoms, there is evidence to suggest that impulsivity and risk behavior increase in adolescents with ADHD compared to children with ADHD. Likewise, it is true that ADHD has been reliably reported more commonly in males (p. 4). Yet, I would appreciate it if the authors could go into more detail about the underlying mechanisms contributing to this gender bias (e.g., different symptom expressions in boys and girls contribute to under-identification of ADHD in girl. The inattentive subtype of ADHD is more prevalent in girls, whereas the combined subtype is more prevalent in boys).

Author response: Thank you for pointing this out. We have elaborated and the revised text reads as follows on p.4:

“Age. Evidence suggested that impulsivity in general reaches its peak during adolescents, which can be explained by brain development [18], however, ADHD symptom presentation changes across development, with declining hyperactivity-impulsivity as age increases, while inattention remains relatively stable [19-22]. Compared to younger children, adolescents develop more maturity in self-regulation and their ability to sustain attention grows [23, 24].”

“Gender. ADHD has been reliably found to be more common in males [33-36]. Different symptom expressions across gender might contribute to under-identification 

of ADHD in females. For example, females with ADHD demonstrated less ADHD core symptoms, less externalising behaviours, but more intellectual impairment and internalising problems [37-39]. Males with ADHD also presented more hyperactive behaviours and more deficits in inhibition and cognitive flexibility, but not attention [40]. In addition, subtype prevalence across gender might also affect identification of ADHD. The inattentive subtype of ADHD is more prevalent in females while the combined subtype is more prevalent in males [3].”

Comment 2: I think it is also important to point out that the existence of a discrepancy between observers says nothing about the accuracy of their respective judgments. Because both informants assess the child's separate environments (home versus school), some of the observed behavioral discrepancy may well reflect true differences in the child's behavior in the two contexts instead of reflecting an informant bias. That is, statements about the accuracy of the respective informants can only be made based on validation with a clinical interview - which, unfortunately, is rarely done in research. I would appreciate if the authors could add a brief overview of this issue in the theory section (e.g., this issue is well presented in the papers by Los Reyes et al., 2015, 2019). Likewise, it should also be mentioned in the discussion that that the present findings do not inform about rater accuracy.

Author response: Thank you for pointing this out and the reference provided. We have elaborated and the revised text reads as follows on p.3 and p.18:

(p.3) “Discrepant results in ADHD ratings might be due to rater effects and 

cross-situation variability on behaviour presentations and demand [9, 15, 16].” 

(p.18) “Nevertheless, although our findings confirm results from the previous literature, the current findings are not informative regarding rater accuracy. Informant discrepancy might reflect differing symptom presentations in different contexts rather than informant bias.” 

Comment 3: On p. 8, the authors present their hypotheses and “predicted that parents with lower education may identify fewer problems, even if the child presents with inattention-hyperactivity symptoms, which will result in discrepancies with teacher reports”. I think, this hypothesis cannot be referred from the present data. Since no clinical interview was additionally performed with the children, information on the actual inattention-hyperactivity symptoms of the children is missing so we do not know if the child presents with respective symptoms or not. The authors should thus rephrase the sentence.

Author response: Thank you for pointing this out. We have rephrased the sentence, and the revised text reads as follows on p. 8:

 “For instance, we predicted that parents with lower education may identify fewer problems than teachers which will result in discrepancies with teacher reports.” 

Comment 4: Although the statistical analysis is appropriate for the question at hand, I wonder whether carrying out a comparable analysis on a latent (instead of on a manifest) level would be possible given the large sample size. (However, I am not an expert in this field, so I am not sure whether those methods exist).

Author response: Thank you for pointing this out. We think this is an interesting suggestion. However, this would be beyond the scope in the current manuscript because that would be beyond the original research question set in the current study. The suggested analysis would be valuable to implement in another manuscript if possible. However, we would be happy to reconsider this issue if the editor or reviewer believe it would be of high value to the journal readership.

 

Comment 5: In the discussion, I would like the authors to elaborate on the effect sizes of the respective characteristics. The discussion summarizes the results well, makes appropriate comparisons with previous research, and addresses possible reasons for inconsistent findings from other studies. What I still miss in the discussion is that the authors elaborate on the effect sizes of the significant results – especially given the large sample. For example, it can be seen from Tables 2 and 3 that some of the significant factors examined only show small effects on parent-teacher agreement, while others show larger effects. In my opinion, adding a few sentences on the relevance of the respective significant characteristics would strengthen the discussion.

Author response: Thank you for pointing this out. We have revised text reads as follows on p. 15-17:

“Regarding child’s characteristics, male children were more likely to fall into parent-teacher disagreement in our study. Male children were found to be 1.93 times more likely to be rated as at risk by parent only, and 3.29 times more likely to be screened positive by teacher only, compared to when both informants agreed not at risk. This is consistent with previous findings for externalizing disorders [e.g., 11] and with previous research about gender features of ADHD where it has classically been found that symptoms and diagnosis are more common in males, whether in single-informant or both-informant ratings [e.g. 77]. However, it is noticeable that parent-only screen-positive was 41.43% less likely than teacher-only and 50.13% less likely than both-informant rating among male children. This may imply the interaction of gender with informant agreement, for example, teacher-informants may have a higher likelihood of overestimating the incidence among male children. 

Regarding parental characteristics, younger parent-informants were more likely to fall into parent-teacher disagreement, with around 2% more likely for each year of parental age. This is consistent with Cheng and colleagues’ [11] finding that younger mother-informants were associated with externalizing problems measured by the SDQ. The current result is consistent with previous studies on younger parental age as a risk factor for informant discrepancy when screening ADHD and other externalizing problems in children [46, 47]. In addition, our results showed that parent-teacher disagreement was more common when parents were screen-positive for facing emotional issues. When parent-informants were screened positive for emotional issues, their children were 62% more likely to be rated by parent only than for both informants to agree not at risk. Harvey et al. [42] reported similar findings regarding the association between maternal depression and mother-teacher disagreement; mothers with depression reported more attention problems in their children than teachers. Our findings were consistent with the depression-distortion hypothesis [53], indicating that parents with psychological distress tended to rate more symptoms and were 40.87% more likely to screen positive alone or 35.03% more likely to agree with teacher that the child screens positive compared to be rated by teacher only.

An association between parental education level and parent-teacher disagreement was observed in our study. When parents did not complete a formal degree, the child was 69% more likely to be screened positive by parent only than for both raters to agree not at risk. This is consistent with previous research that has found a positive correlation between parental education levels and knowledge of ADHD [48, 50] and a positive effect of parental education on ADHD identification [52]. In the current study, children with parents without a formal degree were 49.38% less likely to be screened positive by teacher-only than by both informants.

Regarding family characteristics, some measures of SES, such as number of working parents, and average household income, were significantly associated with parent-teacher disagreement in reporting ADHD symptoms. Children with neither parent working were 61% more likely to be reported by teacher-only compared to when both agreed not at risk. Children with neither parent working were 32.30% less likely to be screened positive by parent-only than by both informants. Children from families with below average income were 33% more likely to be reported by parent only rather than both reporters agreeing not at risk. Our findings suggested that children from families with below average income were rated to have less symptoms by teachers, which contradicts previous findings. For example, in one study of pre-school children [61], both parents and teachers report more symptoms for children with lower SES. This contrast may be due to different measures of SES in the current study (income only) or another unexplained mechanism. Difference in measures of SES may have hinder comparison of current results with previous studies. 

Our findings also suggested that parents who identified their ethnicity as Non-White rated lower hyperactivity than teachers, which is consistent with previous studies [42, 60, 61]. Children with non-White ethnicity were 40% less likely to be screen positive by parent-only compared to both raters agreeing not at risk and 50.82% less likely to be screen positive by parent-only than by teacher-only. Children with non-White ethnicity were 2.44 times more likely to be screened positive by teacher-only than by both informants. Our findings also suggested that children rated by Non-White parent-informants were more likely to be rated as at risk for ADHD by teacher-alone. This finding is similar to a previous study comparing Non-Latino and Latino youths, which found that Latino youths were rated for more symptoms by teachers [62]. However, the results might not be directly comparable, as the classification of ethnicity in this previous study is Latino-oriented. This might suggest the phenomenon appearing across different minor ethnicities. Parental marital status and number of children in the household were not associated with discrepancy in the current study.” 

Thank you again for your valuable feedback. We look forward to hearing from you in due course. 

Sincerely,

Hei Ka (Nadia) Chan

Corresponding Author: Hei Ka Chan (hkchan3@sheffield.ac.uk)

Additional Contact: Richard Rowe (r.rowe@sheffield.ac.uk)

Department of Psychology, 

University of Sheffield

---

## [Decision Letter · Decision Letter 1]

20 Feb 2024

Factors associated with parent-teacher hyperactivity/inattention screening discrepancy: Findings from a UK national sample

PONE-D-23-24488R1

Dear Dr. Chan,

We’re pleased to inform you that your manuscript has been judged scientifically suitable for publication and will be formally accepted for publication once it meets all outstanding technical requirements.

Kind regards,

Gerard Hutchinson, MD

Academic Editor

PLOS ONE

Additional Editor Comments (optional):

Reviewers' comments:

Reviewer's Responses to Questions

**Comments to the Author**

1. If the authors have adequately addressed your comments raised in a previous round of review and you feel that this manuscript is now acceptable for publication, you may indicate that here to bypass the “Comments to the Author” section, enter your conflict of interest statement in the “Confidential to Editor” section, and submit your "Accept" recommendation.

Reviewer #2: All comments have been addressed

2. Is the manuscript technically sound, and do the data support the conclusions?

Reviewer #2: Yes

3. Has the statistical analysis been performed appropriately and rigorously? 

Reviewer #2: Yes

4. Have the authors made all data underlying the findings in their manuscript fully available?

Reviewer #2: Yes

5. Is the manuscript presented in an intelligible fashion and written in standard English?

Reviewer #2: Yes

6. Review Comments to the Author

Reviewer #2: This is an interesting study on a relevant problem faced by clinician in the diagnosis of ADHD in children and adolescents, the pervasiveness of the symptoms. Symptoms of innatention and/or hyperactivity/impulsivity must be reported by parents and other informants, mostly by teachers that normally allocate a subtantial amount of time with children. Based on a large sample of children and adolescents and using validated questionnaires, the study focuses on the parent-teacher agreement of ADHD symptoms bringing new and important findings for discussion. The statistical methods were well conducted and reported, as well the results. The discussion effectively explores the findings, draws relevant comparisons to prior research, and identify potential explanations for discrepancies in results observed in other studies. The authors also responded to the reviewers’ suggestions appropriately.

7. PLOS authors have the option to publish the peer review history of their article (what does this mean?). If published, this will include your full peer review and any attached files.

Reviewer #2: No

---

## [Editor Report · Acceptance letter]

7 May 2024

PONE-D-23-24488R1 

PLOS ONE

Dear Dr. Chan, 

I'm pleased to inform you that your manuscript has been deemed suitable for publication in PLOS ONE. Congratulations! Your manuscript is now being handed over to our production team.

Kind regards, 

on behalf of

Dr. Gerard Hutchinson 

Academic Editor

PLOS ONE